# The 2024 Annual Meeting of the Essential Programmes on Immunization Managers in Central Africa: A Peer Learning Platform

**DOI:** 10.3390/vaccines13030301

**Published:** 2025-03-11

**Authors:** Franck Mboussou, Audry Mulumba, Celestin Traore, Florence Conteh-Nordman, Shalom Tchokfe Ndoula, Antoinette Demian Mbailamen, Jean Claude Bizimana, Christian Akani, Yolande Vuo-Masembe, Bridget Farham, Marcelin Menguo Nimpa, Thomas Noel Gaha, Martin Morand, Lynda Rey, Maria Carolina Danovaro-Holliday, Charles Shey Wiysonge, Benido Impouma

**Affiliations:** 1World Health Organization Regional Office for Africa, Brazzaville P.O. Box 06, Congosheyc@who.int (C.S.W.);; 2Essential Immunization Programme, Ministry of Health, 30 Avenue de la Justice, Kinshasa M7MG+2X8, Democratic Republic of the Congo; 3UNICEF Regional Office for West and Central Africa, Dakar P.O. Box 29720, Senegal; 4Gavi Secretariat, 1218 Le Grand-Saconnex, Switzerlandlrey@gavi.org (L.R.); 5Essential Immunization Programme, Ministry of Health, 8 Rue 3038 Quartier du Lac, Yaoundé P.O. Box 1937, Cameroon; 6Essential Immunization Programme, Ministry of Health, Ndjamena P.O. Box 548, Chad; 7Essential Immunization Programme, Ministry of Health, Rue Pierre Ngendandumwe, Bujumbura J968+H3J, Burundi; 8World Health Organization, Democratic Republic of Congo Country Office, 42 Avenue des Cliniques, Kinshasa 01206, Democratic Republic of the Congo; 9UNICEF, Democratic Republic of Congo Country Office, 372 Concession Immotex, Kinshasa M7H9+HQW, Democratic Republic of the Congo; 10World Health Organization Headquarters, 1211 Geneva, Switzerland

**Keywords:** immunization, Central Africa, big catch-up, peer learning

## Abstract

Background: Since 1974, Essential Programme on Immunisation managers from ten Central African countries meet yearly with partners to review progress made and share experiences and lessons learned from the implementation of immunization programmes. The 2024 meeting occurred in Kinshasa, Democratic Republic of Congo, in September 2024. This conference report summarizes the key takeaways from discussions on using immunization data for decision-making, the implementation of the Big Catch-Up (BCU) initiative to reduce the burden of zero-dose children, and progress and challenges in introducing selected new vaccines. Conference Takeaways: Inaccurate administrative data on routine immunization observed in most countries, compared to WHO/UNICEF Estimates of National Immunization Coverage and national survey estimates, affect timely decisions to improve the Expanded Programme on Immunization (EPI) performance. Five countries in Central Africa are among the priority countries of the BCU initiative but, as of the end of August 2024, are yet to formally start its implementation. Cameroon and Central African Republic introduced the malaria vaccine in January 2024 and August 2024, respectively, while the Democratic Republic of Congo, Chad, and Burundi have planned to do so by 2025. Conclusions and Recommendations: Meeting participants put forward several recommendations for countries and immunization partners, including but not limited to (i) investing more in routine immunization data quality assurance to better use data to inform decisions, (ii) accelerating the implementation of the BCU initiative to close the immunity gap resulting from routine immunization disruptions due to the COVID-19 pandemic, (iii) updating malaria vaccine introduction plans to invest more in demand generation and community engagement, and (iv) learning from Cameroon’s experience in tackling hesitancy to human papilloma virus vaccine. It is critical to set up an appropriate mechanism for monitoring the implementation of these recommendations.

## 1. Introduction

In 1974, the World Health Organization (WHO) established the Expanded Programme on Immunization to ensure equitable access to life-saving vaccines for every child, regardless of geographic location or socioeconomic status [1]. Later, this programme evolved into the Essential Programme on Immunization (EPI), seeking to safeguard individuals of all ages through comprehensive immunization efforts [1]. Countries across the world gradually set up the EPI at the national level. In the Central Africa subregion made up of 10 countries (Angola, Burundi, Cameroon, Central African Republic (CAR), Chad, Congo, Democratic Republic of the Congo, Equatorial Guinea, Gabon, and São Tomé and Príncipe), the EPI was operational in all countries by 1984. As a result of the implementation of the EPI in Central Africa, (i) the number of antigens provided to children and adolescents increased from 6 in 1984 to 13 in 2024; (ii) the number of children vaccinated yearly with the first dose of the pentavalent vaccine—Diphtheria, Tetanus, Pertussis, *Haemophilus influenzae* type b, and Hepatitis B—(Penta 1) increased from 1.6 million in 2000 to 7.9 million in 2023; and (iii) the median coverage with the third dose of the pentavalent vaccine (Penta 3) increased from 38.5% [range: 31%;82%] in 2000 to 72% [range: 42%;89%] in 2023, as per the 2024 revision of the WHO and UNICEF Estimates of National Immunization Coverage (WUENIC) [2]. Despite this significant and commendable progress, most countries in Central Africa are still off-track towards achieving global immunization agenda 2030 (IA2030) targets [3]. Since 1974, EPI managers from Central African countries meet every year with partners and other stakeholders to review progress made and share experiences and lessons learned from implementing national immunization programmes in line with global and regional goals. Organized jointly by the WHO Regional Office for Africa and UNICEF Regional Office for West and Central Africa, the 2024 annual EPI managers’ meeting took place in Kinshasa, Democratic Republic of Congo, from 9 to 12 September 2024 and attracted 157 participants from 10 countries of the subregion and immunization partners. The meeting’s methods included oral presentations from subject matter experts, panel discussions on selected topics, and country posters on EPI progress. This report summarizes the key takeaways from discussions on three key topics: use of immunization data for decision-making, implementation of the big catch-up (BCU) initiative aiming at reducing the burden of zero-dose and under immunized children [3], and progress and challenges in the introduction of selected new vaccines (Malaria vaccines and Human Papillomavirus vaccines).

## 2. Conference Section

### 2.1. Routine Immunization Data Use to Inform Decisions

Discussion surrounding the use of routine vaccination coverage data to inform decision-making showed that all countries in Central Africa have set up a system for collecting data on vaccine uptake and computing immunization coverage to monitor EPI progress. At health facility level, the system involves paper-based data collection/reporting tools, which include registers, tally sheets, coverage wall charst, monthly reporting forms, etc. In 6 of the 10 countries (Angola, Burundi, Cameroon, Chad, Congo, and Democratic Republic of Congo), monthly reports from health facilities are submitted to the regional and national level through an electronic tool for aggregated data reporting, i.e., District Health Information Software 2 (DHIS2). The remaining four countries use a Microsoft Excel-based tool for data transmission from the district level to the national level: the district vaccination management data tool (DVMDT) and are in the process of transitioning to DHIS 2. In all countries, target populations for EPI are derived from census projections. Suboptimal data quality assurance processes and inaccurate estimates of target populations, including outdated censuses or incomplete civil registration and vital statistics (CRVSs), have led to overestimating immunization coverage in some countries. Of the 10 countries in Central Africa, administrative coverage with Penta 3 in 2023 was similar to WUENIC estimates in 5 countries (Burundi, Congo, Equatorial Guinea, Gabon, and São Tomé and Príncipe). In Angola, Cameroon, Central African Republic, Chad, and the Democratic Republic of Congo, the administrative penta 3 coverage in 2023 was more significant than WUENIC, suggesting the underestimation of target populations, the overreporting of children vaccinated, or a combination of both (Figure 1). In these countries, the proportion of districts with administrative penta 3 coverage in 2023 of at least 80% was 51% in Angola, 54% in Cameroon, 80% in Central African Republic and Chad, and 85% in the Democratic Republic of Congo, suggesting disparities in the EPI performance between districts.

Based on WUENIC estimates, the median Penta 1 to Penta 3 dropout rate in 2023 was 6%, with four countries recording over 10% (Democratic Republic of Congo (20%), Chad (17%), Angola (15%) and Central African Republic (12%)) and three countries recording less than 5% (Burundi (0%), Gabon (3%) and Congo (4%)).

While WUENIC data are used to monitor and compare the performance of immunization programmes at the national and international levels in a standardized way [4], microplanning and actions aiming to improve the routine immunization performance are undertaken using administrative coverage data. Immunization coverage surveys (ICSs) are among the data sources used to generate WUENIC estimates in addition to administrative data and other programmatic and operational considerations. In 2023–2024, seven countries conducted an ICS: Cameroon, Central African Republic, Chad, Congo, the Democratic Republic of Congo, Equatorial Guinea, and São Tomé and Príncipe. The Democratic Republic of Congo also employed a Demographic and Health Survey (DHS) in 2023–2024, as did Angola. Limited technical capacities, financial resources (especially when a sample representative of districts is required), and low vaccination card availability were identified as the main challenges of ICSs in Central Africa. None of the Central African countries have managed to deploy an electronic immunization registry nationwide, failing to leverage the COVID-19 vaccination individual data systems to strengthen routine immunization data management and use for decision-making [5].

Inaccurate administrative data, especially the overestimation of immunization coverage, affects timely and well-informed decision-making aiming to improve EPI performance. The following recommendations were adopted to enhance effective use of quality data for decision-making: (i) conduct data triangulation exercises annually to adjust population size estimates using WHO methods for Assessing and Improving the Accuracy of Target Population Estimates for Immunization Coverage [6]; (ii) invest more in data quality self-assessments and audits to improve routine immunization data accuracy; (iii) consider deploying electronic immunization registries, including digital vaccination cards learning from the experience of countries in other regions, and leveraging COVID-19 vaccination experience; and (iv) conduct immunization coverage surveys representative of the district level more frequently.

### 2.2. Efforts to Reduce the Burden of Zero-Dose Children

During the meeting, a discussion panel was dedicated to countries’ efforts in reducing the burden of zero-dose and under-immunized children. Zero-dose refers to children who have failed to receive any routine vaccination, while under-immunized refers to children who have received some, but not all, of their recommended scheduled vaccinations. In all Central African countries, COVID-19 disruptions have led to a dramatic increase in the number of zero-dose and under-immunized children. From 2019 to 2023, based on WUENIC (2023 revision) and the United Nations Population Division (UNPD) estimates (2024 revision), Central Africa recorded an estimated 13.9 million of zero-dose and under-immunized children (8.9 million zero-dose children and 5.0 million under-immunized children). Table 1 presents the distribution of the estimated number of zero-dose and under-immunized children by country in Central Africa from 2019 to 2023.

Angola, Chad, Cameroon, and the Democratic Republic of Congo account for 91% of zero-dose and under-immunized children in the subregion.

To close the immunity gaps caused by the backsliding of immunization coverage during the COVID-19 pandemic and catch up on vaccination of 90% of zero-dose children by the end of 2025, WHO, UNICEF, Gavi Secretariat and Gates Foundation set up the BCU initiative in 2023 [7]. Five Central African countries are among the priority countries for this initiative: Burundi, Cameroon, Chad, Central African Republic, and Democratic Republic of Congo. Having transitioned out of Gavi support in 2024, Angola is not part of the priority countries but has engaged in BCU activities. As of the end of August 2024, as part of the BCU initiative preparedness, the priority five countries had revised their immunization schedules to extend vaccination targets to children aged 24–59 months and allow delayed vaccination. However, at the time of this meeting, the revised schedules were yet to be officially validated and disseminated within the health system in Burundi and Central African Republic. In addition, vaccination data collection tools were revised and deployed to capture vaccine doses given to children aged 24–59 months in three countries: Cameroon, Chad, and the Democratic Republic of Congo. In Burundi and the Central African Republic, the tools have been revised but not yet distributed in health facilities. As of the end of August 2024, Gavi had approved a total of 21 million vaccine doses for the five BCU priority countries , including 10.3 million doses already shipped in four countries (49%), with Burundi yet to receive the first shipment.

As of the end of August 2024, none of the five countries have started to formally implement BCU activities, particularly the periodic intensification of routine immunization (PIRI) campaigns to catch up on vaccination of children missed since 2020. Cameron has scheduled three rounds of BCU PIRI in September and December 2024 and January 2025. In Chad, the first round of the BCU PIRI was scheduled for December 2024 in the 23 regions that make up the country. Burundi had planned four rounds of BCU PIRIs separated by four weeks, with the first round scheduled in late December 2024. In Central African Republic and the Democratic Republic of Congo, BCU PIRIs are planned for the first quarter of 2025. Angola, a country that transitioned out of Gavi support in 2024, benefited from Gavi funding through the Middle-Income Countries Approach (MICs) to catch up on the vaccination of zero-dose children. With USD 9.1 million received out of USD 43 million required, MIC funding had already contributed to reducing the zero-dose burden by 21% at the time of the meeting. The funding gap for BCU in Angola was yet to be covered. Wrapping up this discussion, participants adopted the following recommendations: (i) Burundi and Central African Republic are to ensure that revised schedules and tools to accommodate delayed vaccination are distributed to all health facilities, (ii) the five BCU priority countries are to make all necessary arrangements to conduct the first PIRI rounds by end January 2025, and (iii) international partners are to support the Government of Angola to raise funds through public–private partnership to cover the BCU funding gap.

### 2.3. Introduction of New Vaccines

Discussion on progress in new vaccine introduction focused on the rollout of malaria vaccines and progress in revitalization of HPV vaccination.

Vaccination has been recently added to the malaria prevention package, following prequalification by the World Health Organization (WHO) of two vaccines: RTS/AS01, approved in 2022 [8], and R21/Matrix-M, approved in 2023 [9]. As of the end of August 2024, two countries have started to roll out malaria vaccines in Central Africa: Cameroon and Central African Republic. Cameroon started to administer RTS/AS01 vaccine on 22 January 2024 in 42 districts at most risk of malaria of the 200 health districts in the country [10]. The recommended schedule in Cameroon comprises four doses that are administered to children at 6, 7, 9, and 24 months of age. By end July 2024, i.e., six months into the vaccine rollout, 74 401 children were vaccinated with the first malaria vaccine dose, providing 50% coverage, against 26% for the second dose and 7% for the third dose. Cameroon conducted a mini post-introduction evaluation of malaria vaccine in May–June 2024. The following causes of low vaccination coverage and high dropout rate were identified: (i) the fact that vaccination schedules for the first and second doses do not match existing contacts with immunization services as per the official schedule, (ii) insufficient investment in demand generation, outreach and mobile vaccination sessions, and a robust defaulters’ tracking system, and (iii) suboptimal use of the missed opportunities for vaccination approach. Central African Republic started to roll out R21/Matrix-M vaccine on 22 August 2025 in all the 35 districts of the country. The recommended schedule in Central African Republic comprises four doses administered to children at 6, 7, 9, and 16 months of age. The initial results of malaria vaccine rollout in Central African Republic were not available at the time of the EPI managers meeting. Three other countries had planned to roll out malaria vaccine in the last quarter of 2024 or the first quarter of 2025: Burundi (first quarter 2025), Chad (October 2024), and Democratic Republic of Congo (October 2024).

In December 2022, the Gavi Board approved the revitalization of the Alliance’s Human Papillomavirus (HPV) vaccine programme with the aim of reaching over 86 million girls with HPV vaccine by 2025 to avert over 1.4 million future deaths from cervical cancer [11]. In Central Africa, apart from Cameroon and São Tomé and Príncipe, who pioneered the introduction of HPV vaccine in Central Africa in 2020 and 2021, respectively, no other Gavi eligible country has started to roll out these vaccines.

Cameroon started to administer HPV vaccines to girls aged 9–13 years using a school-based implementation strategy while dealing with COVID-19 vaccination, resulting in low uptake. The coverage in girls was 18.2% in 2021 and 20.3% in 2022. Concerns about vaccine safety, fear of infertility and the negative influence of social media exacerbated by the COVID-19 pandemic are known to be the main reasons for HPV vaccine hesitancy in Cameroon [12]. In 2022, WHO updated recommendations on HPV vaccine, adding an alternative single dose schedule. As a follow-up, Cameroon switched to a single-dose schedule and extended the target population to boys from the same age group in 2023. As a result, the HPV vaccine coverage in girls increased to 56.8% in 2023 compared to 20.3% in 2022. The coverage in boys in 2023 was 26%. Lessons learned from Cameroon underline the need for innovative communication, political commitment, and advocacy targeting faith leaders to sustain and expand vaccination efforts.

São Tomé and Príncipe has been less impacted by HPV vaccine hesitancy, despite the context of the COVID-19 pandemic, using a two-dose schedule. In 2022, the HPV coverage in girls aged 9–14 years was 74% with the first dose and 55% with the second dose, compared to 80% with the first dose and 58% with the second dose in 2023.

In addition to Cameroon and São Tomé and Príncipe, two other Gavi-eligible countries have planned to introduce HPV vaccine: Burundi in 2025 and the Democratic Republic of Congo in 2026. 

Among the three Gavi non-eligible countries in Central Africa, Equatorial Guinea implemented an HPV vaccine demonstration project in 2024 in one district, achieving 80.3% coverage of girls aged 9–13. Leveraging lessons from the demonstration project, the Equatorial Guinea Government has planned a nationwide rollout by 2026. Angola has planned to introduce the HPV vaccine to girls aged 9–12 in 2025, as 1.4 million vaccine doses out of 2.2 million needed to reach 2.13 million girls have already been delivered. Gabon is contemplating starting a demonstration project in the coming years. It is critical for Burundi, Democratic Republic of Congo, Equatorial Guinea, and Angola to learn from Cameroon and São Tomé and Príncipe’s experience on schedules, primary and secondary target groups, and response to vaccine hesitancy to ensure successful introduction of HPV vaccine.

Table 2 summarizes the status of malaria and HPV vaccines in Central African countries as of the end of August 2025.

## 3. Conclusions

The 2024 EPI managers meeting in Central Africa offered an opportunity to share experiences and devise solutions. It allowed for the exploration of ways to deal better with the common issue of data quality in routine immunization to inform timely decisions. The meeting was also a successful peer learning platform for EPI managers and other stakeholders in immunization programmes on the topics of data use for decision-making, BCU readiness and implementation and introduction of malaria and HPV vaccines, among other topics not covered in this report.

The discussions during the meeting highlighted immunization data quality issues in most countries, leading to overestimation of immunization coverage and preventing timely decision-making. Participants recommended that countries invest more in routine immunization data quality assurance for a better use of data to inform decisions.

The high burden of zero-dose children resulting from routine immunization disruptions during the emergency phase of the COVID-19 pandemic was discussed. The implementation of the BCU Initiative sponsored by Gavi in five priority countries, among those with the highest number of zero-dose children, has been delayed mainly due to competing priorities. In Angola, a country that transitioned out of Gavi support in 2024, the implementation of catch-up vaccination activities was slowed down for financial reasons. Given the urgent need for reducing immunity gaps to prevent resurgence of long-forgotten vaccine preventable diseases, it is critical to accelerate the implementation of the BCU initiative in Gavi eligible countries and to support Angola’s efforts in raising local funding for catch-up vaccinations through public–private partnership.

The first six months of the malaria vaccine rollout in Cameroon have shown suboptimal coverage for the first dose and a high dropout rate for the second and the third doses. A mini PIE conducted four months into the introduction showed that the vaccination schedule with appointments for the first and second doses not matching existing contacts with health facilities for vaccination, and that insufficient investment in demand generation and outreach/mobile vaccination sessions, are the main drivers of low coverage and children’s retention rate. Central African Republic and the three other countries that will introduce malaria vaccines in the fourth quarter of 2024 or the first quarter of 2025 were urged to update their introduction plan to ensure enough investment in demand generation and community engagement to allow time for mothers to become familiar with new appointments related to malaria vaccines.

HPV vaccine introduction in Cameroon was hampered by high vaccine hesitancy, contributing to low acceptance. Switching from a two-dose to a single-dose schedule and extending the target population to boys contributed to improved coverage in girls. It is critical for countries planning to introduce HPV vaccine to learn from Cameroon’s experience in tackling vaccine hesitancy and contemplate a single dose schedule.

Lessons learned and recommendations from the meeting, if implemented, will ultimately contribute to improving EPI’s performance in these programme areas. Indeed, peer learning has been shown to play an important role in rapidly advancing technical capacities to drive policy change and address issues that affect vaccination [13].

It is critically important and urgent that the meeting organizers set up an appropriate mechanism for monitoring the implementation of the meeting’s recommendations and document best practices. Adding a page on the meeting website, displaying the action points for EPIs by country and for partners, and allowing online monthly updates, coupled with regular meetings to monitor progress in implementing action points, would increase the chance of turning the meeting recommendations into concrete actions. EPI managers are expected to lead the required changes for more efficient and effective immunization programmes. To this end, international partners are urged to set up a training programme on transformational leadership for EPI managers and key EPI staff. The participants agreed that the 2025 EPI managers’ meeting will occur in São Tomé and Príncipe.

### Limitations

This paper is a meeting report. The content was developed using oral presentations and panel discussion summaries. The situation may have evolved since the meeting’s conclusion, so the interpretation of the results presented here should take into account these limitations.

## Figures and Tables

**Figure 1 vaccines-13-00301-f001:**
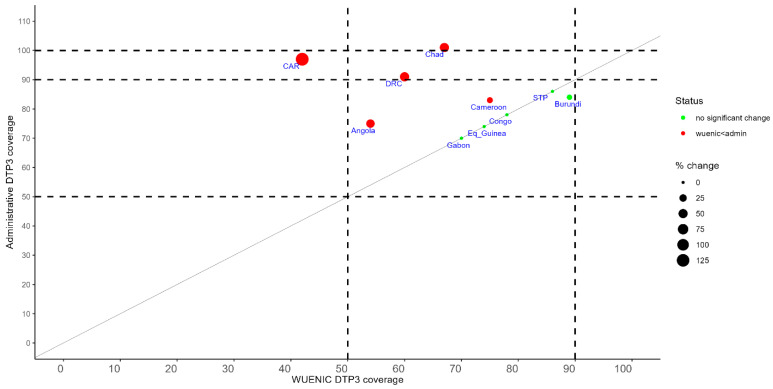
WUENIC and administrative coverage with the third dose of pentavalent vaccine in 2023 in Central Africa. Dashed lines earmark 50%, 80%, and 90% coverage; the solid line represents the diagonal.

**Table 1 vaccines-13-00301-t001:** Distribution of zero-dose and under-immunized children in Central Africa (2019–2023).

Country	Surviving Infants (2019–2023 Cohorts)	# Zero-Dose Children	# Under-Immunized Children	# Zero-Dose and Under-Immunized Children	% of All Zero-Dose and Under-Immunized Children in Central Africa
Angola	6,389,000	2,468,000	744,000	3,212,000	23
Burundi	2,200,000	119,000	57,000	176,000	1
Cameroon	4,503,000	1,088,000	243,000	1,331,000	10
Central African Republic	1,054,000	485,000	127,000	612,000	4
Chad	3,683,000	818,000	743,000	1,561,000	11
Congo	899,000	172,000	34,000	206,000	1
Democratic Republic of Congo	19,920,000	3,636,000	3,050,000	6,686,000	48
Equatorial Guinea	258,000	59,000	25,000	84,000	1
Gabon	334,000	93,000	15,000	108,000	1
São Tomé and Príncipe	31,000	2000	-	2000	0
**Total**	**39,271,000**	**8,940,000**	**5,038,000**	**13,978,000**	**100**

#: Number

**Table 2 vaccines-13-00301-t002:** Status of introduction of malaria and human papilloma virus vaccines in Central African countries as of the end of August 2024.

Country	Malaria Vaccine Introduction	HPV Vaccine Introduction
Angola	Not yet planned	Planned in 2025
Burundi	Planned in 2025	Planned in 2025
Cameroon	Introduced in January 2024	Introduced in 2020
Chad	Planned in October 2024	Not yet planned
Central African Republic	Planned in September 2025	Not yet Planned
Congo	Not yet planned	Not yet planned
Democratic Republic of Congo	Planned in October 2024	Planned in 2026
Equatorial Guinea	Demonstration in 2024; introduction planned in 2026	Not yet planned
Gabon	Not yet planned	Not yet planned
São Tomé and Príncipe	Not yet planned	Introduced in 2021

## Data Availability

All the presentations and materials from the 2024 EPI Managers meeting in Central Africa are available and downloadable at the following link: https://who-ist-ca.github.io/Reunion-Annuelle-PEV/epi/agenda.html (accessed on 7 March 2025).

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
