# Peer review of "The 2024 Annual Meeting of the Essential Programmes on Immunization Managers in Central Africa: A Peer Learning Platform"

_vaccines, 2025, doi:10.3390/vaccines13030301_

Round 1
Reviewer 1 Report
Comments and Suggestions for Authors
The paper is well presented. Only minor editing may be needed. In line 45-47: "...10 countries (Angola, Burundi, Cameroon, Central African 45 Republic (CAR), Chad, Congo, Democratic Republic of the Congo, Equatorial Guinea, Gabon, and Sao Tome and Principe, the EPI...". Should the comma be changed to a parenthesis?
Author Response
Comment 1: The paper is well presented. Only minor editing may be needed. In line 45-47: "...10 countries (Angola, Burundi, Cameroon, Central African 45 Republic (CAR), Chad, Congo, Democratic Republic of the Congo, Equatorial Guinea, Gabon, and Sao Tome and Principe, the EPI...". Should the comma be changed to a parenthesis?
Response 1: Thank you for this suggestion. This has been corrected (see line 53)
Reviewer 2 Report
Comments and Suggestions for Authors
Abstract
Line 33. I suggest adding the key progress and take away
Line 181. Introduction of new vaccines
Comment: it will be interesting to provide a table describing the events in addition to narrative, as it is much easier to grasp the progress
Line 236. Conclusions. I recommend structuring the conclusions to provide the significant hurdles and the way ahead for each of the research goals.
Author Response
Comment 1: Line 33. I suggest adding the key progress and take away
Response 1: Thank you for this comment. We have added a summary of key progress and takeaways in the introduction section to provide a clearer overview of the conference outcomes (see lines 34-40).
Comment 2: Line 181. Introduction of new vaccines. it will be interesting to provide a table describing the events in addition to narrative, as it is much easier to grasp the progress
Response 2: Thank you for this suggestion. A table summarizing the status of malaria and HPV vaccine introduction in Central African countries as of end august 2024 has been added (see Table II, line 260)
Comment 3: Line 236. Conclusions. I recommend structuring the conclusions to provide the significant hurdles and the way ahead for each of the research goals.
Response 3: Thank you for this suggestion. We have restructured the conclusion taking into account this suggestion (see lines 270-300)
Reviewer 3 Report
Comments and Suggestions for Authors
Dear Authors,
the manuscript is well written and interesting:
Line 29: explain the acronym EPI
Line 47: close the parenthesis
Line 78: in "six" countries (not five)
Line 81: The remaining "four" countries (not five)
Line 88: in "four" countries (not five). Which group does Gabon belong to?
Line 125: in other countries in general (why only Americas region?)
Rewrite the Table 1 title header better
Author Response
Comment 1: The manuscript is well written and interesting: Line 29: explain the acronym EPI
Response 1: Thank you for this suggestion. This has addressed (see line 29)
Comment 2: Line 47: close the parenthesis.
Response 2: Thank you for this comment. The parenthesis has been closed as suggested (see line 53).
Comment 3: Line 78: in "six" countries (not five)
Response 3: Thank you for this comment. This has been corrected (see line 84).
Comment 4: Line 81: The remaining "four" countries (not five)
Response 4: Thank you for this comment. This has been corrected (see line 87).
Comment 5: Line 88: in "four" countries (not five). Which group does Gabon belong to?
Response 5: Thank you for this comment. Indeed, it’s five countries including Gabon. This has been corrected (see line 95).
Comment 6: Line 125: in other countries in general (why only Americas region?)
Response 6: Thank you for this comment. We have revised the text to broaden the scope of learning to other regions, not just the Americas (see line 134).
Comment 7: Rewrite the Table 1 title header better
Response 7: Thank for this comment. Table I title has been corrected (see line 151).
Reviewer 4 Report
Comments and Suggestions for Authors
I think this is a very important meeting report. It is detailed and well written.
However, there needs to be more focus and emphasis on detailing both reasons/ causes for the gaps in immunization coverage as well as the factors that led to success in a couple of countries.
Also, while the report states that monitoring needs to be set up and stepped up - it would have been more important to understand if the " how" of monitoring was discussed. Otherwise, the next meeting will have the same issues.
There are many lessons in EPI that can be learned from other countries and adapted - there should be a sense of urgency and therefore this report is important
Author Response
Comment 1. I think this is a very important meeting report. It is detailed and well written. However, there needs to be more focus and emphasis on detailing both reasons/ causes for the gaps in immunization coverage as well as the factors that led to success in a couple of countries.
Response 1: Thank you for the appreciation and for the comment related to causes of gaps in immunization coverage and success of countries. The reasons behind irregular Immunization Coverage surveys were specified in lines 119-121. In the same line, underestimation of target populations or overreporting of children vaccinated or a combination of both, were identified as causes of discrepancies between WUENIC and administrative coverages (lines 126-128). The reasons for delayed implementation of the Big Catch-Up initiative in some countries were described (see lines 185-190). With regards to Malaria vaccine introduction, the mini Post-Introduction Evaluation results in Cameroon highlighted the causes of low coverage with dose 1 and high dropout rate (see lines 206-211). The same apply to HPV (see line 228-231). The analysis in this report focused on comparing WUENIC and Administrative coverages and not on countries’ performance in immunization coverage. We have added in the conclusion section, a recommendation on documentation of best practices (see line 318).
Comment 2: Also, while the report states that monitoring needs to be set up and stepped up - it would have been more important to understand if the " how" of monitoring was discussed. Otherwise, the next meeting will have the same issues.
Response 2: Thank you for this suggestion. This has been addressed in the conclusion section (see lines 318-321).
Comment 3: There are many lessons in EPI that can be learned from other countries and adapted - there should be a sense of urgency and therefore this report is important.
Response 3: Thank you for this comment. We have specified that implementing action points form the meeting as well as documenting best practices is critically important ang urgent (see lines 316).
Round 2
Reviewer 4 Report
Comments and Suggestions for Authors
Thank you for editing it so well. You have explained the importance of demand generation and addressing vaccine hesitancy in your conclusion section - and these are important points.
It is now much clearer and complete as a report that can be shared with other countries to learn lessons from.